# Modelling the Effects of Roselle Extract, Potato Peel Flour, and Beef Fat on the Sensory Properties and Heterocyclic Amines Formation of Beef Patties Studied by Using Response Surface Methodology [note 1]

**DOI:** 10.3390/foods10061184

**Published:** 2021-05-25

**Authors:** Anna Judith Pérez-Báez, Martin Valenzuela-Melendres, Juan Pedro Camou, Gustavo González-Aguilar, Orlando Tortoledo-Ortiz, Humberto González-Ríos, Manuel Viuda-Martos

**Affiliations:** 1Centro de Investigación en Alimentación y Desarrollo, A.C. Carretera Gustavo E. Astiazarán Rosas, #46. Hermosillo, Sonora 83304, Mexico; anna.perez@estudiantes.ciad.mx (A.J.P.-B.); jpc@ciad.mx (J.P.C.); gustavo@ciad.mx (G.G.-A.); otortoledo@ciad.mx (O.T.-O.); hugory@ciad.mx (H.G.-R.); 2Grupo de Industrialización de Productos de Origen Animal, Departamento de Tecnología Agroalimentaria, Escuela Politécnica Superior de Orihuela, Universidad Miguel Hernández, Carretera Beniel, Km 3.2, E-Orihuela, 03312 Alicante, Spain

**Keywords:** heterocyclic amines, functional meat products, roselle, potato peel, beef patties

## Abstract

Heterocyclic amines (HCAs) are compounds with carcinogenic potential formed during high-temperature processing of meat and meat products. Vegetables or their extracts with high antioxidant capacity can be incorporated into the meat matrix to reduce their formation, but it is necessary to find the optimal levels to achieve maximum inhibition without affecting the sensory properties. The objective of this study was to evaluate the effects of roselle extract (RE, 0–1%), potato peel flour (PP, 0–2%), and beef fat (BF, 0–15%) on the sensory properties and formation of HCAs in beef patties using response surface methodology. IQx, IQ, MeIQx, MeIQ, 4,8-DiMeIQx, and PhIP were identified and quantified by HPLC. Regression models were developed to predict sensory properties and HCAs’ formation. All models were significant (*p* < 0.05) and showed a R^2^ > 0.70. Roselle extract and beef fat had a negative linear effect on the formation of the total HCAs, while PP had a positive linear effect. The optimal formula that minimizes the formation of HCAs included 0.63% RE, 0.99% PP, and 11.96% BF. RE and PP are foods that can be used as ingredients in low-fat beef patties to minimize the formation of HCAs without affecting their sensory properties.

## 1. Introduction

Several epidemiological studies have shown that diet can be an important factor in the incidence of various types of cancer. A higher incidence of cancer has been reported when the intake of fruits and vegetables is low [1], when the consumption of foods with an elevated glycemic index increase [2], or when the meat and meat processed intake is high [3,4]. Recently, processed meat was classified as a carcinogen (Group 1) and red meat as a probable carcinogen (Group 2A) by the International Agency for Research in Cancer [5]). The World Cancer Research Fund [6] reported strong evidence that intake of meat and processed meat is a convincing cause of colorectal cancer. Huang et al. [7] reported a strong relationship between red and processed meat consumption and multiple cancer types. Their study revealed that an increment of 100 g/d of red meat and 50 g/d of processed meat consumption was associated with 11–51% and 8–72% higher risk of multiple cancer outcomes.

Processing meat and meat products at high temperatures can generate genotoxic compounds such as heterocyclic amines. These compounds are formed from Maillard reactions between creatine, free amino acids, and residual glucose present in meat [8]. The HCAs ingested through foods and subsequently absorbed into the body are converted into genotoxic metabolites mediated by the N-acetyltransferase 1 (NAT1), N-acetyltransferase 2 (NAT2), hepatic cytochrome P-450 1A2 (CYP1A2), and sulfotransferases enzymes [9]. These metabolites react with the DNA of various tissues to form adducts that lead to cancer problems. The HCAs have shown mutagenic capacity in the Ames test [10] and carcinogenic capacity in studies with rodents [11,12]. Besides, it has been demonstrated the formation of HCAs-DNA adducts in colorectal human tissue [13]. This evidence suggest that the HCAs play an important role in the cancer incidence in frequent meat consumers.

The increase of the risk of cancer due to ingestion of HCAs from food is an important issue in which is necessary to propose solutions, either by reducing the formation of these compounds during meat preparation or by limiting their absorption and activation when the food is ingested. Some strategies to decrease HCAs during food preparation are temperature and cooking time reduction [14], applying spice-based marinades [15], adding antioxidants [16], or reducing fat content in meat products [17]. Beef fat is an ingredient with important technological advantages because it contributes to the flavor, texture, and juiciness of meat products [18]. However, the current trend is to reduce their consumption because saturated fat from red meat has been directly related to the risk of developing many cancer types, including advanced prostate cancer [19], breast cancer [20], and ovarian cancer [21]. Moreover, beef fat is susceptible to oxidation, which is accelerated during cooking, producing free radicals that may be involved in the formation of HCAs. 

The use of vegetable tissues and extracts as an antioxidant source is one of the best strategies to reduce HCAs’ formation during the cooking of meat products [16]. Roselle extract and potato peel are potential ingredients in meat product formulations with the capacity to reduce HCAs’ formation. Roselle extracts exhibit high reducing power and free radicals scavenging [22], highly effective in reducing HCAs’ formation in food [15]. However, the incorporation of roselle extract decreases the sensory properties in meat products [23]. On the other hand, potato peel is a co-product generated by potato processing industries, usually discarded or used as animal feed [24]. Potato peel is rich in dietary fiber [25], exhibits high antioxidant capacity [26], improves water-holding capacity, shelf life, and sensory characteristics of meat products [27]. Moreover, potato peel has the potential to decrease HCAs’ formation in beef burgers [28] and to absorb HCAs during the digestion of foods [29]. According to the evidence, incorporating roselle extract and potato peel flour into the meat formulation could reduce HCAs formation, but it is possible that high levels of these ingredients can affect the sensory properties of the product. For this reason, is important to optimize the incorporation of roselle extract and potato peel flour into beef patties formulation to have a product with lower quantities of HCAs, with no effect on sensory properties.

When non-traditional ingredients are incorporated, or the traditional ones are reduced during the development of new meat products, it is necessary to evaluate their possible effects and interactions to optimize their inclusion without affecting product quality. Response surface methodology (RSM) is a mathematical tool used to optimize the incorporation of ingredients during the development of meat products [30]. The objective of this research was to evaluate the effects of the addition of roselle extract, potato peel flour, and beef fat on the formation of HCAs in beef patties using RSM and optimize its incorporation without affecting their sensory properties. 

## 2. Materials and Methods

### 2.1. Raw Materials

Lean beef, inside round (Semimembranosus, 74% moisture, 6% lipids, 18.5% protein, and 1.5% ash), pH 5.85 and fat (34% moisture, 63% lipids, 2% protein, and 1% ash), pH 6.34 were obtained from a local market and used on the same day of purchase. The meat was cut into pieces no larger than 5 cm^3^ and ground using a Hobart grinder (Hobart model 4152, Troy, OH, USA) through a 4.7 mm plate. Beef fat was ground in the same way as meat. Ground meat and fat were stored at 2 °C until further use. 

Roselle (*Hibiscus sabdariffa* L.) was purchased in a local market, and it was dried in a convection oven (ENVIRO-PAK, model Micro-Pak, series MP500, Clackamas, OR, USA) at 60 °C for 6 h. Roselle extract (RE) was prepared by hot-water extraction of roselle petals and subsequent freeze-drying to 8% moisture, as specified by Perez-Baez et al. [30]. Potato peels were dried to 5% moisture and ground [30].

### 2.2. Reagents

Unless otherwise stated, all reagents were HPLC, analytical grade or purchased from Sigma-Aldrich (St. Louis, MO, USA). Sodium hydroxide (NaOH), hydro-chloric acid (HCl), and ammonium hydroxide were analytical grade. Methanol (MeOH), ethyl acetate, formic acid, and acetonitrile were HPLC grade. For the determination of heterocyclic amines (HCAs), reagents from Toronto Research Chemicals (Downsview, Ontario, Canada) were purchased: 2-amino-3-methylimidazo [4,5-f]quinoline (IQ), 2-amino-3-methylimidazo[4,5-f]quinoxaline (IQx), 2-amino-3,4-dimethylimidazo[4,5-f]quinoline (MeIQ), 2-amino-3,4,8-trimethyl-3H-imidazo[4,5-f]quinoxaline (MeIQx), 2-amino-3,4,8-trimethylimidazo[4,5-f]quinoxaline (4,8-DiMeIQx), 2-amino-1-methyl-6-phenylimidazo[4,5-b]pyridine (PhIP), and 2-amino-3,4,7,8-tetramethylimidazo[4,5-f]quinoxaline (4,7,8-TriMeIQx) were used as internal standard.

### 2.3. Experimental Design

The effect of the incorporation of roselle extract (RE, 0–1%), potato peel flour (PP, 0–2%), and beef fat (BF, 0–15%) on the formation of HCAs (IQx, IQ, MeIQx, MeIQ, 4,8-DiMeIQx, PhIP), and sensory properties (flavor, texture, and juiciness) of beef patties were studied. The formation of HCAs and sensory properties were evaluated using a response surface model with a central composite design (CCD) with three factors (RE, PP and BF). Twenty experimental runs in random order, with six replications on the central point and simple runs for the rest of the treatments of the CCD, were performed (Table 1). Two repetitions of the complete experiment were carried out, and in each repetition, three samples were analyzed per treatment for each evaluation.

### 2.4. Beef Patties Preparation 

Twenty different treatments of beef patties were prepared according to the experimental design shown in Table 1. For the preparation of beef patties, 2 kg batches were used for each treatment. The ingredients for each treatment were homogenized in a manual mixer (LEM Products, West Chester, OH, USA) for 3 min, and the beef patties were shaped using a manual patty forming machine (9 cm diameter × 1 cm thickness) to obtain 70 g per unit. After forming, beef patties were immediately cooked using a grilling unit (George Foreman, model GR2121P, Miramar, FL, USA), 1 min each side and then 15 s per side until reaching 71 °C, measured in the geometrical center with a thermocouple.

### 2.5. Sensory Analysis 

Sensory evaluation of patties was performed by twenty-eight panelists, consisting of 15 females and 13 males, experienced in sensory evaluation and consumed beef patties regularly. The panel was comprised of students and staff of the Department of Food Technology of Animal Origin. The sensorial evaluation took place in an environmentally controlled (21 ± 1 °C, 55 ± 5% relative humidity) room partitioned into booths. 

Two separate training sessions were held before evaluation, during which beef patties samples manufactured for this study were used. Patties were cooked as specified in Section 2.4 and maintained warm at 60 °C in an oven until testing. Beef patties were cut into 2 cm^3^ cubes, placed in hermetic containers coded with a 3-digit number and presented randomly to panelists. Water and unsalted crackers were offered to panelists for palate cleansing. Panelists were asked to indicate their score on a 10 cm line scale ranging from 0 to 10. Each point was converted to a numerical value. Flavor (0 = not intense, 10 = intense), and texture (0 = soft, 10 = firm), and juiciness (0 = dry, 10 = moist) were scored.

### 2.6. Extraction of Heterocyclic Amines 

Heterocyclic amines were extracted following the methodology described by Messner and Murkovic [31] with slight modifications. Briefly, one gram of freeze-dried sample was dissolved in 12 mL of 1 mol/L NaOH and homogenized for 1 h at 300 rpm at room temperature. This suspension was mixed with 13 g of diatomaceous earth (Extrelut NT, Merck, Darmstadt, Germany), homogenized, and loaded into a column of Extrelut NT20. Extraction was performed by washing the column with 75 mL of ethyl acetate, which was passed through a Waters Oasis MCX cartridge (3 mL, 60 mg). Next, the cartridges were sequentially rinsed with 2 mL of 0.1 mol/L HCl and 2 mL methanol. The retained HCAs were eluted with 2 mL of an ammonia hydroxide-methanol (19:1, *v/v*) mixture. The samples were evaporated to dryness under nitrogen and dissolved in 100 µL of methanol for further quantification.

### 2.7. Identification and Quantification of Heterocyclic Amines by HPLC-DAD 

The identification and quantification of HCAs were conducted on an HPLC 1220 Infinity instrument (Agilent Technologies, LC, Santa Clara, CA, USA) equipped with a diode-array detector (Agilent, DAD, Santa Clara, CA, USA). The separation was performed on a reversed-phase analytical column, TSK-GEL ODS-80TM 5 µm (4.6 × 25 cm) from Tosoh Bioscience (Labkemi, Lund, Sweden) protected by a guard column (LC-18-DB, Stockholm, Sweden) at 35 °C. The following mobile phases were used: A, methanol/acetonitrile/acetic acid/water (8:14:2:76, *v*/*v*/*v*/*v*) pH 5; B, acetonitrile, with an injection volume of 10 µL of standard and extracts and an elution flow rate of 0.7 mL/min. The gradient program started with 100% A (0–10 min); 77% A at 20 min; hold 77% A at 30 min; 100% at 45 min. 

The identification of HCAs was carried out, comparing the retention time recorded by each standard and samples spiked with HCAs standards at 264 nm. The recovery rate of HCAs in the samples was determined by adding standards before extraction. The concentration of the HCAs in the samples was calculated using a standard curve following the indications of Haskaraca et al. [32]. Quantitative determination was performed using an external calibration curve method. A linear regression of amine concentration (ng) versus the peak area of the amines was performed. The regression coefficients (R^2^) for the amines were: 0.997, 0.996, 0.958, 0.985, 0.994, and 0.978 for IQx, IQ, MeIQx, MeIQ, 4,8-DiMeIQx, and PhIP, respectively.

### 2.8. Statistical Analysis 

After performing the total experimental runs specified in Table 1, a quadratic model was fitted to the response values using Design Expert software (V.7.6.1, Stat-Ease, Inc., Minneapolis, MN, USA). Prediction models were developed using response surface methodology (RSM), according to the following generalized second-order polynomial equation:(1)$$y=\beta_{0}+\overset{3}{\underset{i=1}{\sum }}\beta_{i}X_{1}+\overset{3}{\underset{i=1}{\sum }}\beta_{ii}X_{i}^{2}+\underset{i<j}{\sum }\beta_{ij}X_{i}X_{j}+\varepsilon \;\;$$
where *y* are the response variables (sensory analysis and formation of heterocyclic amines); *β*_0_, *β_i_*, *β_ii_*, and *β_ij_* are the intercept, linear, quadratic, and interaction regression coefficients, respectively; *X_i-j_* are independent variables (roselle extract, potato peel flour, and beef fat). The analysis of variance (ANOVA) and the determination coefficient (R^2^) of the models were determinate.

The optimal incorporation of RE, PP, and BF of beef patties was predicted by selecting the desired targets for each response variable. Heterocyclic amines (IQ, IQx, MeIQ, MeIQx, 4,8-DiMeIQx, and PhIP) were kept as minimum targets, while sensory properties were kept as maximum targets.

## 3. Results and Discussion

### 3.1. Sensory Analysis

Table 2 shows the experimental values for the sensory analysis. Flavor, texture, and juiciness scores ranged from 6.8 to 7.5, 6.4 to 7.6, and 6.0 to 7.7, respectively. Consequently, the values obtained for each of the treatments are considered within the acceptable ranges for each sensory attribute [33]. 

Table 3 shows the average and regression models for sensory properties prediction as a function of roselle extract, potato peel flour, and beef fat. In addition, the analysis of variance, *F*-value, and the coefficients of determination (R^2^) of the regression model.

The models were considered adequate for response variable prediction when they showed a *p*-value < 0.05 and R^2^ values greater than 0.80 [34]. Figure 1 shows the effect and tendencies of the process variables on the sensory parameters studied. The flavor was affected by the incorporation of potato peel flour with a negative linear effect, while roselle extract and fat had no significant effect. According to the model (Figure 1), the incorporation of potato peel flour into the beef patty formulation decreased flavor of the product. Marconato et al. [34] found comparable results for beef patties with sweet potato peel flour (1.5%). The negative effect of potato peel flour on beef patty flavor is attributed to the bitter flavor of hydroxycinnamic acid derivatives, mainly chlorogenic acid [35]. 

On the other hand, the addition of roselle extract in the beef patties formulation decreased the texture and juiciness of the product, while beef fat increased these properties as shown by a negative and positive linear effect, respectively (Table 3). It has been reported that the incorporation of roselle extract to meat products decreases the pH, inducing denaturation of myofibrillar proteins, and decreases water and fat retention capacities [36], affecting the texture and juiciness of the product. The negative effect of roselle extract on juiciness was reverted by the incorporation of potato peel flour, as indicated by the positive interaction between these two factors. The analysis of the sensory properties is fundamental during the development of new meat products since they are attributes that are closely related to consumer preference and food acceptance [37]. This is why designing prediction models that allow visualizing the behavior of the sensory properties evaluated is important.

The use of non-conventional ingredients such as roselle extract and potato peel flour, or the reduction of traditional ingredients such as fat, affects the sensory properties of beef patties, as evidenced in the present study. However, these ingredients also have the ability to modify the formation of heterocyclic amines, and it is important to consider this property during the optimization process.

### 3.2. Heterocyclic Amine Analysis

The values for HCAs are shown in Table 2. Individual HCAs contents varied from 0 to 18.09 ng/g for IQx, from 0.20 to 11.66 ng/g for IQ, from 0.29 to 3.93 ng/g for MeIQx, from 0.04 to 11.51 ng/g for MeIQ, from 0.65 to 1.72 ng/g for 4,8-DiMeIQx and from 1.87 to 15.18 ng/g for PhIP. The sum of the six studied HCAs were considered as total amines, and the values ranged from 4.3 ng/g to 51.8 ng/g. Table 3 shows the means and regression equations obtained for the prediction of the heterocyclic amines formed during the cooking of beef patties prepared with roselle extract, potato peel flour, and fat. Figure 2 shows the response surface graphics in 3D for total HCAs, based on roselle extract addition, potato peel flour, and beef fat. The regression models for the HCAs values were significant (*p* < 0.05) and had a determination coefficient higher than 0.7.

Linear equations (*p* < 0.05) were adjusted in the formation of IQ and 4,8-DiMeIQx, obtaining a significate decrease by the roselle extract and a positive linear effect by potato peel flour. While the polynomial equations of IQx, MeIQx, MeIQ, and PhIP (*p* < 0.05) were obtained, observing a negative linear effect by the roselle extract and a positive linear effect by potato peel flour addition. Total HCAs content was affected by roselle extract and beef fat with negative linear effects and by potato peel flour with a positive linear effect.

According to the models specified in Table 3, the incorporation of roselle extract decreases HCAs content. The observed values for total HCAs in the beef patties added with roselle extracts are similar to some reported studies, where the effect of other vegetable extracts was evaluated [17,32,38]. Thus, Lu et al. [12] reported that the total HCAs content in cooked pork patties with partial replacement of fat by olive oil and cooked at 180 and 220 °C were 4.11 and 20.03 ng/g, respectively, while the samples where the fat was replaced by sunflower oil the values obtained were 5.98 and 23.88 ng/g, respectively. However, Oz and Cakmak [38] informed that the total HCAs found in beef meatballs added with conjugated linoleic acid in the ratios of 0.05%, 0.1%, 0.25%, and 0.5% (*w*/*w*) and grilled at 150, 200 and 250 °C ranged between 0.03 and 1.49 ng/g.

The addition of ingredients with potential antioxidant activity to the meat matrices will reduce the formation of HCAs. In this way, Vitaglione and Fogliano [39] mentioned that the antioxidant inhibitory properties of several compounds might be due to the consequence of diverse actions interfering at different stages of the HCAs formation. Roselle extracts, due to the high anthocyanins content, mainly delphinidin-3-*O*-glucoside, delphinidin-3-*O*-sambubioside, and cyanidin-3-*O*-sambubioside [40], have shown a high antioxidant activity [41,42]. It has been shown that antioxidants decrease the HCAs formation due to antioxidants stabilizing the pyrazine and/or pyrazyl radicals avoiding the union with creatinine, and this way the aminoimidazole part of the amine is not formed [43].

On the other hand, the potato peel flour incorporation had a contrary effect to what was expected, increasing the total HCAs content. We expect that the phenolic acids and flavonoids found in potato peel flours (chlorogenic and caffeic acids) could contribute to reducing the HCAs formation in beef patties, as indicated by Zeng et al. [44]. However, it is very important to notice that the exact mechanism of the inhibitory effect of polyphenolic compounds (phenolic acids or flavonoids including anthocyanins) on the total quantity of HCAs remains poorly understood. A possible explication on the positive effect of potato peel flour on the total HCAs is that this ingredient contains a carbohydrate quantity of 275–1014 equivalents of glucose per gram of peel [45], producing an increase in glucose, which is a precursor of the HCAs formation, and causing a greater amount of total HCAs.

Conversely, the beef fat content in treatments has a negative linear effect on the total HCAs content. According to the model, increasing fat content in beef patties formulation, decreasing HCAs content. Similar results were reported by Hwang and Ngadi [46] in meat emulsions at different fat contents. The negative effect of fat addition on the formation of HCAs could be related to a decrease in protein levels due to an increase in fat content that influences free amino acids and creatine concentration, which are the basic substrates for HCAs formation. This hypothesis is supported by Szterk and Jesionkowska [47] when compared the HCAs content on culinary elements of red meat with different fat contents (roast beef < sirloin < rib eye). According to the results of this study, rib eye had the lowest amount of precursors indispensable for HCAs synthesis because this culinary element, in comparison to roast beef and sirloin, contained the lowest amount of proteins resulted from a higher level of fat.

Models generated in this study could be used to predict the HCAs formation as a function of roselle extract, potato peel flour, and beef fat contents in beef patties. However, to optimize the incorporation of these ingredients in the formulation, we must not only minimize the formation of HCAs, but it is also important not to affect the sensory properties of the product. 

### 3.3. Optimization

The numerical optimization tool of the Design Expert software was used to estimate the optimal combination of roselle extract, potato peel flour and beef fat that minimize HCAs formation and maximize sensory properties of beef patties. The optimal formulation was obtained when maximal desirability was reached. The resulting optimal formulation included 0.63% of roselle extract, 0.99% of potato peel, and 11.96% of beef fat, and the obtained desirability was 0.7. Under these optimal conditions, the predicted values for the HCAs formation and sensory properties were: IQx = 0.0 ng, IQ = 2.49 ng, MeIQx = 0.64 ng, MeIQ = 0.13 ng, 4,8-DiMeIQx = 1.15 ng, and PhIP = 1.87 ng, flavor = 7.3, texture = 7.5, and juiciness = 7.5.

## 4. Conclusions

The incorporation of roselle extract decreases HCAs’ formation, but it causes a decrease in the levels of acceptance of texture and juiciness in beef patties, while the potato peel flour has a contrary effect. The incorporation of fat improves the sensorial properties and has a linear negative effect on the formation of amines. The optimized formula that minimizes HCAs’ formation and maximizes sensory properties included 0.63% of RE, 0.99% of PP, and 11.96% of BF. Roselle extract, potato peel flour, and fat can be used as optimized ingredients to formulate beef patties with low heterocyclic amines formation without affecting sensory properties.

## Figures and Tables

**Figure 1 foods-10-01184-f001:**
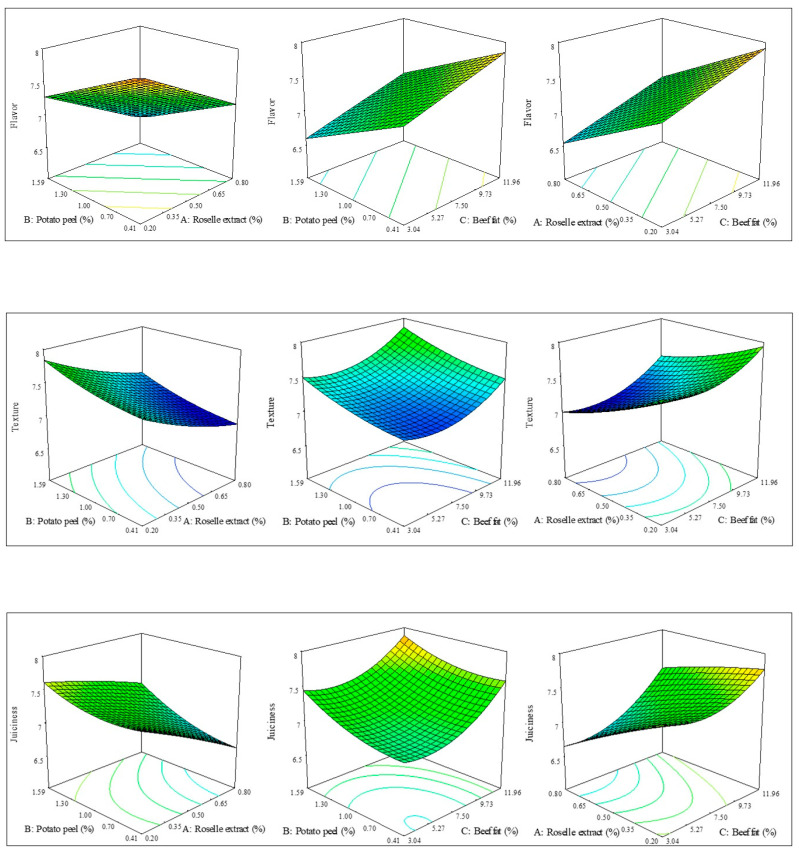
The effects of roselle extract, potato peel, and beef fat on the estimated flavor, texture, and juiciness of beef patties.

**Figure 2 foods-10-01184-f002:**
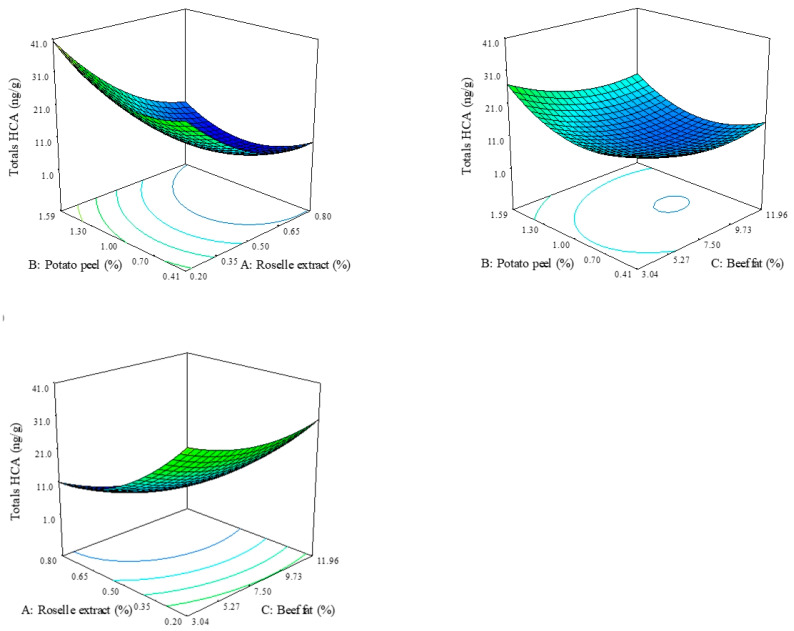
The effects of roselle extract, potato peel flour, and beef fat on the estimated total heterocyclic amines of beef patties.

**Table 1 foods-10-01184-t001:** Experimental combinations of treatments of beef patties with added roselle extract, potato peel flour and beef fat.

Run ^a^	Coded Values			Experimental Values		
	X_1_	X_2_	X_3_	RE	PP	BF
1	−1	−1	−1	0.20	0.40	3.04
2	1	−1	−1	0.80	0.40	3.04
3	−1	1	−1	0.20	1.60	3.04
4	−1	−1	1	0.20	0.40	11.96
5	1	1	−1	0.80	1.60	3. 04
6	1	−1	1	0.80	0.40	11.96
7	−1	1	1	0.20	1.60	11.96
8	1	1	1	0.80	1.60	11.96
9	−1.68	0	0	0	1.00	7.50
10	1.68	0	0	1.00	1.00	7.50
11	0	−1.68	0	0.50	0	7.50
12	0	1.68	0	0.50	2.00	7.50
13	0	0	−1.68	0.50	1.00	0
14	0	0	1.68	0.50	1.00	15.00
15	0	0	0	0.50	1.00	7.50
16	0	0	0	0.50	1.00	7.50
17	0	0	0	0.50	1.00	7.50
18	0	0	0	0.50	1.00	7.50
19	0	0	0	0.50	1.00	7.50
20	0	0	0	0.50	1.00	7.50

^a^ Runs 1–8 are factorial points; 9–14 are axial points; 15–20 are central points. X_1_ = roselle extract (RE, %), X_2_ = potato peel flour (PP, %), and X_3_ = beef fat (BF, %).

**Table 2 foods-10-01184-t002:** Experimental response for sensory properties and heterocyclic amines of beef patties with added roselle extract, potato peel flour, and beef fat.

Run	Sensory Properties				Heterocyclic Amines (ng/g)					
	Flavor	Texture	Juiciness	IQx	IQ	MeIQx	MeIQ	DiMeIQx	PhIP	Totals
1	7.8 ± 0.8	7.7 ± 1.2	7.4 ± 0.8	10.1 ± 5.9	5.5 ± 2.6	3.9 ± 0.4	1.9 ± 0.7	1.2 ± 0.1	14.1 ± 0.5	36.8 ± 5.8
2	6.5 ± 1.5	7.0 ± 1.0	6.3 ± 1.6	0.1 ± 0.0	3.0 ± 0.1	2.2 ± 0.2	0.2 ± 0.0	0.7 ± 0.2	10.8 ± 0.8	17.0 ± 0.9
3	7.4 ± 1.4	8.0 ± 0.7	7.7 ± 1.5	10.6 ± 1.4	11.7 ± 0.4	3.8 ± 0.1	11.5 ± 1.1	1.5 ± 0.5	12.7 ± 0.4	51.8 ± 1.7
4	6.2 ± 1.3	7.2 ± 0.9	7.1 ± 1.1	4.6 ± 0.6	6.2 ± 0.2	2.0 ± 0.2	1.0 ± 0.7	0.8 ± 0.1	6.9 ± 1.4	21.5 ± 0.3
5	8.3 ± 1.0	7.8 ± 0.7	7.8 ± 1.3	2.7 ± 0.0	6.7 ± 0.5	3.1 ± 0.4	6.2 ± 0.1	1.7 ± 0.4	14.0 ± 1.4	34.4 ± 1.1
6	7.3 ± 0.9	7.2 ± 0.7	7.4 ± 1.1	1.6 ± 0.1	1.7 ± 0.1	0.3 ± 0.1	1.4 ± 0.1	0.8 ± 0.1	4.6 ± 0.8	10.5 ± 1.2
7	7.8 ± 0.8	8.0 ± 0.8	8.1 ± 1.4	18.1 ± 0.0	7.2 ± 0.2	2.4 ± 0.3	8.0 ± 0.5	1.7 ± 0.2	8.3 ± 1.4	45.7 ± 1.1
8	7.0 ± 0.8	7.5 ± 0.9	7.8 ± 1.1	2.2 ± 0.3	1.5 ± 0.3	2.7 ± 0.3	1.3 ± 0.3	1.3 ± 0.0	2.4 ± 0.3	11.2 ± 0.9
9	7.8 ± 1.1	7.9 ± 1.1	7.6 ± 1.6	9.2 ± 0.3	6.9 ± 0.3	1.7 ± 0.1	8.4 ± 0.2	1.4 ± 0.4	15.2 ± 0.2	42.8 ± 0.0
10	6.9 ± 1.3	7.0 ± 1.0	6.6 ± 1.2	0.2 ± 0.1	0.2 ± 0.0	0.4 ± 0.3	1.0 ± 0.4	0.7 ± 0.0	1.9 ± 0.3	4.3 ± 0.1
11	8.3 ± 1.2	7.1 ± 1.1	7.2 ± 1.2	3.3 ± 0.2	2.6 ± 0.3	3.8 ± 0.1	4.6 ± 0.7	1.3 ± 0.4	8.8 ± 0.7	24.3 ± 1.4
12	6.2 ± 1.5	7.9 ± 1.2	7.7 ± 1.3	13.5 ± 0.8	4.2 ± 0.1	0.7 ± 0.2	5.6 ± 0.2	1.0 ± 0.1	2.1 ± 0.4	27.1 ± 2.9
13	7.0 ± 1.5	7.3 ± 0.8	7.6 ± 1.0	7.7 ± 1.8	4.8 ± 1.1	2.9 ± 0.4	2.5 ± 1.5	0.7 ± 0.1	4.6 ± 0.5	23.1 ± 1.0
14	7.6 ± 0.7	8.4 ± 1.0	8.0 ± 1.2	3.3 ± 0.3	3.3 ± 0.6	1.3 ± 0.2	1.0 ± 0.2	1.3 ± 0.3	2.3 ± 0.2	12.6 ± 1.3
15	7.1 ± 1.7	7.3 ± 1.1	7.0 ± 1.6	0.4 ± 0.1	4.5 ± 0.5	1.8 ± 0.3	0.5 ± 0.0	1.1 ± 0.1	4.6 ± 0.4	12.8 ± 0.7
16	7.1 ± 1.2	7.2 ± 1.1	7.0 ± 1.3	0.4 ± 0.1	3.9 ± 0.0	0.3 ± 0.1	1.0 ± 0.3	1.5 ± 0.1	5.3 ± 0.3	12.4 ± 0.6
17	7.2 ± 1.0	7.4 ± 0.7	7.2 ± 0.8	0.0 ± 0.0	2.7 ± 0.1	0.5 ± 0.0	0.3 ± 0.0	1.0 ± 0.3	2.5 ± 0.3	6.9 ± 0.6
18	6.9 ± 0.9	7.3 ± 0.8	7.2 ± 0.7	0.0 ± 0.0	4.6 ± 2.2	0.4 ± 0.0	0.8 ± 0.1	1.4 ± 0.2	4.3 ± 0.3	11.6 ± 2.6
19	6.9 ± 1.3	7.2 ± 0.5	7.1 ± 1.2	0.0 ± 0.0	2.7 ± 0.5	0.3 ± 0.0	0.9 ± 0.0	0.8 ± 0.3	5.2 ± 1.0	9.7 ± 0.3
20	7.4 ± 1.2	7.1 ± 1.2	7.2 ± 1.9	0.0 ± 0.0	2.5 ± 0.1	0.3 ± 0.0	0.0 ± 0.0	0.8 ± 0.2	4.3 ± 0.4	8.0 ± 0.0

Values are the means *±* SD from experimental determinations. Sensory properties were evaluated using a 10 cm line scale, where flavor, texture and juiciness were ranked from 0 = not intense/soft/dry to 10 = intense/firm/moist.

**Table 3 foods-10-01184-t003:** Analysis of variance of the regression models and regression coefficients for sensory properties and heterocyclic amines formed during cooking of beef patties prepared with roselle extract, potato peel flour, and beef fat.

Response	Model	Means ± SD	R^2 a^	*F*-Value	Prob > F	Polynomial Equation ^b^
Sensory properties						
Flavor	linear	7.23 ± 0.29	0.80	21.07	0.0001	7.23 − 0.42A − 0.36B * + 0.26C
Texture	Quadratic	7.47 ± 0.17	0.90	9.78	0.0007	7.24 − 0.30A * + 0.18B + 0.17C * − 0.02AB + 0.03AC + 0.01BC + 0.06A^2^ * + 0.06B^2^ + 0.20C^2^
Juiciness	Quadratic	7.35 ± 0.15	0.95	18.81	0.0001	7.12 − 0.30A * + 0.19B + 0.26C + 0.08AB * + 0.11AC − 0.04BC *– 0.02A^2^ * + 0.12B^2^ + 0.24C^2^ *
Heterocyclic amines						
IQx	Quadratic	4.40 ± 2.39	0.90	9.40	0.0008	0.14 − 3.52A * + 2.79B * − 0.60C − 1.36AB − 0.12AC + 1.38BC + 1.55A^2^ * + 2.86B^2^ * + 1.83C^2^ *
IQ	Linear	4.32 ± 1.57	0.70	11.60	0.0003	4.32 − 2.19A * + 0.90B * − 0.85C
MeIQx	Quadratic	1.74 ± 0.94	0.73	3.07	0.048	0.56 − 0.61A * − 0.28B − 0.45C + 0.37AB + 0.13AC + 0.25BC + 0.32A^2^ + 0.73B^2^ * + 0.69C^2^ *
MeIQ	Quadratic	2.90 ± 1.22	0.93	14.38	0.0001	0.59 − 2.65A * + 1.01B * − 0.03C − 1.34AB * + 0.09AC − 1.09BC * + 1.43A^2^ * + 1.57B^2^ * + 0.40C^2^
DiMeIQx	Linear	1.14 ± 0.23	0.66	10.08	0.0006	1.14 − 0.28A * + 0.03B + 0.18C *
PhIP	Quadratic	6.74 ± 2.31	0.86	6.59	0.003	4.26 − 3.43A * − 1.79B * − 1.39C * + 0.11AB − 0.77 AC − 0.34BC + 2.18A^2^ * + 1.07B^2^ + 0.37C^2^
Totals	Quadratic	21.23 ± 4.16	0.95	23.33	0.0001	10.07 − 12.68A * + 2.65B * − 3.15C * –2.64AB − 1.02AC − 0.95BC + 5.87A^2^ * + 6.62B^2^ * + 3.85C^2^ *

^a^ 0 < R^2^ < 1, close to 1 means more significant. ^b^ A: roselle extract, B: potato peel flour, and C: beef fat * *p* < 0.05. SD = standard deviation shown.

## Data Availability

The data presented in this study are available on request from the corresponding author.

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
