# Peer review of "Modelling the Effects of Roselle Extract, Potato Peel Flour, and Beef Fat on the Sensory Properties and Heterocyclic Amines Formation of Beef Patties Studied by Using Response Surface Methodology†"

_foods, 2021, doi:10.3390/foods10061184_

Round 1

Reviewer 1 Report

The effect of roselle extract and potato peel flour on HCA content of beef has been already reported by the authors in a paper with (almost) identical title (Proceedings 70, no. 1: 27. https://doi.org/10.3390/foods_2020-07656; also a MDPI publication). This must be mentioned in the paper. Data provided on HCA content are identical in both papers, and results of a sensory analysis have been added. Thus, the Title should be modified by adding “… on sensory properties."

Introduction: I suggest to include more recent scientific assessments on the effect of red meat consumption on cancer (e.g. by IARC)

Materials and Methods: I understand that the extract was prepared from Rosella leaves.

Sensory: Are the values given in Table means from the scores of 18 panellists? Was there any interaction between panellists (which may cause bias)? What was the range and SD?

Reviewer 2 Report

The work is interesting and original. However, there is no hypothesis in the work. It would be worthwhile to enter the pH value of the meat in the "Raw materials" chapter, as it could have influenced the results. It would be worthwhile to estimate the relationship between sensory quality and HCA content. The effect of beef fat supplementation on HCA formation is poorly discussed. This effect is also the weakening of the effect of temperature on the formation of HCA by fat, as the fat absorbs heat.

Author Response

Please see th attachment

Round 2

Reviewer 1 Report

Line 4: I suggest keeping the last words of the Title as in the previous version (…”studied by using Response Surface Methodology). “Beef Patties using Response Surface Methodology” sounds a bit odd.  

Line 14: I understand that only the HCA data have been presented, if so, the word “partially” should be added here

Line 41: perhaps better “intake of meat and processed meat”

Lines 110ff: I would prefer giving a very brief account on the preparation methods, e.g. “was prepared by hot-water extraction of roselle petals and subsequent freeze-drying to 8% moisture, as specified by Perez-Baez et al [30]. … Potato peels were dried to 5% moisture and ground [30].

Line 402: Reference to the publication by Szterk and Jesionkowska is missing from the reference list

Line 464: Fund, W.C.R. should be “World cancer research fund”
